# Deep content-based music recommendation

**Aäron van den Oord, Sander Dieleman, Benjamin Schrauwen**

Electronics and Information Systems department (ELIS), Ghent University

{aaron.vandenoord, sander.dieleman, benjamin.schrauwen}@ugent.be

## Abstract

Automatic music recommendation has become an increasingly relevant problem in recent years, since a lot of music is now sold and consumed digitally. Most recommender systems rely on collaborative filtering. However, this approach suffers from the cold start problem: it fails when no usage data is available, so it is not effective for recommending new and unpopular songs. In this paper, we propose to use a latent factor model for recommendation, and predict the latent factors from music audio when they cannot be obtained from usage data. We compare a traditional approach using a bag-of-words representation of the audio signals with deep convolutional neural networks, and evaluate the predictions quantitatively and qualitatively on the Million Song Dataset. We show that using predicted latent factors produces sensible recommendations, despite the fact that there is a large semantic gap between the characteristics of a song that affect user preference and the corresponding audio signal. We also show that recent advances in deep learning translate very well to the music recommendation setting, with deep convolutional neural networks significantly outperforming the traditional approach.

## 1 Introduction

In recent years, the music industry has shifted more and more towards digital distribution through online music stores and streaming services such as iTunes, Spotify, Grooveshark and Google Play. As a result, automatic music recommendation has become an increasingly relevant problem: it allows listeners to discover new music that matches their tastes, and enables online music stores to target their wares to the right audience.

Although recommender systems have been studied extensively, the problem of music recommendation in particular is complicated by the sheer variety of different styles and genres, as well as social and geographic factors that influence listener preferences. The number of different items that can be recommended is very large, especially when recommending individual songs. This number can be reduced by recommending albums or artists instead, but this is not always compatible with the intended use of the system (e.g. automatic playlist generation), and it disregards the fact that the repertoire of an artist is rarely homogenous: listeners may enjoy particular songs more than others.

Many recommender systems rely on usage patterns: the combinations of items that users have consumed or rated provide information about the users' preferences, and how the items relate to each other. This is the *collaborative filtering* approach. Another approach is to predict user preferences from item content and metadata.

The consensus is that collaborative filtering will generally outperform content-based recommendation [1]. However, it is only applicable when usage data is available. Collaborative filtering suffers from the *cold start problem*: new items that have not been consumed before cannot be recommended. Additionally, items that are only of interest to a niche audience are more difficult to recommend because usage data is scarce. In many domains, and especially in music, they comprise the majority of

| | Artists with positive values | Artists with negative values |
|---|---|---|
| 1 | Justin Bieber, Alicia Keys, Maroon 5, John Mayer, Michael Bublé | The Kills, Interpol, Man Man, Beirut, the bird and the bee |
| 2 | Bonobo, Flying Lotus, Cut Copy, Chromeo, Boys Noize | Shinedown, Rise Against, Avenged Sevenfold, Nickelback, Flyleaf |
| 3 | Phoenix, Crystal Castles, Muse, Röyksopp, Paramore | Traveling Wilburys, Cat Stevens, Creedence Clearwater Revival, Van Halen, The Police |

Table 1: Artists whose tracks have very positive and very negative values for three latent factors. The factors seem to discriminate between different styles, such as indie rock, electronic music and classic rock.

the available items, because the users' consumption patterns follow a power law [2]. Content-based recommendation is not affected by these issues.

## 1.1 Content-based music recommendation

Music can be recommended based on available metadata: information such as the artist, album and year of release is usually known. Unfortunately this will lead to predictable recommendations. For example, recommending songs by artists that the user is known to enjoy is not particularly useful.

One can also attempt to recommend music that is perceptually similar to what the user has previously listened to, by measuring the similarity between audio signals [3, 4]. This approach requires the definition of a suitable similarity metric. Such metrics are often defined ad hoc, based on prior knowledge about music audio, and as a result they are not necessarily optimal for the task of music recommendation. Because of this, some researchers have used user preference data to tune similarity metrics [5, 6].

## 1.2 Collaborative filtering

Collaborative filtering methods can be neighborhood-based or model-based [7]. The former methods rely on a similarity measure between users or items: they recommend items consumed by other users with similar preferences, or similar items to the ones that the user has already consumed. Model-based methods on the other hand attempt to model latent characteristics of the users and items, which are usually represented as vectors of latent factors. Latent factor models have been very popular ever since their effectiveness was demonstrated for movie recommendation in the Netflix Prize [8].

## 1.3 The semantic gap in music

Latent factor vectors form a compact description of the different facets of users' tastes, and the corresponding characteristics of the items. To demonstrate this, we computed latent factors for a small set of usage data, and listed some artists whose songs have very positive and very negative values for each factor in Table 1. This representation is quite versatile and can be used for other applications besides recommendation, as we will show later (see Section 5.1). Since usage data is scarce for many songs, it is often impossible to reliably estimate these factor vectors. Therefore it would be useful to be able to predict them from music audio content.

There is a large *semantic gap* between the characteristics of a song that affect user preference, and the corresponding audio signal. Extracting high-level properties such as genre, mood, instrumentation and lyrical themes from audio signals requires powerful models that are capable of capturing the complex hierarchical structure of music. Additionally, some properties are impossible to obtain from audio signals alone, such as the popularity of the artist, their reputation and and their location.

Researchers in the domain of *music information retrieval* (MIR) concern themselves with extracting these high-level properties from music. They have grown to rely on a particular set of engineered audio features, such as mel-frequency cepstral coefficients (MFCCs), which are used as input to simple classifiers or regressors, such as SVMs and linear regression [9]. Recently this traditional approach has been challenged by some authors who have applied deep neural networks to MIR problems [10, 11, 12].

In this paper, we strive to bridge the semantic gap in music by training deep convolutional neural networks to predict latent factors from music audio. We evaluate our approach on an industrial-scale dataset with audio excerpts of over 380,000 songs, and compare it with a more conventional approach using a bag-of-words feature representation for each song. We assess to what extent it is possible to extract characteristics that affect user preference directly from audio signals, and evaluate the predictions from our models in a music recommendation setting.

## 2 The dataset

The Million Song Dataset (MSD) [13] is a collection of metadata and precomputed audio features for one million contemporary songs. Several other datasets linked to the MSD are also available, featuring lyrics, cover songs, tags and user listening data. This makes the dataset suitable for a wide range of different music information retrieval tasks. Two linked datasets are of interest for our experiments:

- The Echo Nest Taste Profile Subset provides play counts for over 380,000 songs in the MSD, gathered from 1 million users. The dataset was used in the Million Song Dataset challenge [14] last year.
- The Last.fm dataset provides tags for over 500,000 songs.

Traditionally, research in music information retrieval (MIR) on large-scale datasets was limited to industry, because large collections of music audio cannot be published easily due to licensing issues. The authors of the MSD circumvented these issues by providing precomputed features instead of raw audio. Unfortunately, the audio features provided with the MSD are of limited use, and the process by which they were obtained is not very well documented. The feature set was extended by Rauber et al. [15], but the absence of raw audio data, or at least a mid-level representation, is still an issue. However, we were able to attain 29 second audio clips for over 99% of the dataset from 7digital.com.

Due to its size, the MSD allows for the music recommendation problem to be studied in a more realistic setting than was previously possible. It is also worth noting that the Taste Profile Subset is one of the largest collaborative filtering datasets that are publicly available today.

## 3 Weighted matrix factorization

The Taste Profile Subset contains play counts per song and per user, which is a form of *implicit feedback*. We know how many times the users have listened to each of the songs in the dataset, but they have not explicitly rated them. However, we can assume that users will probably listen to songs more often if they enjoy them. If a user has never listened to a song, this can have many causes: for example, they might not be aware of it, or they might expect not to enjoy it. This setting is not compatible with traditional matrix factorization algorithms, which are aimed at predicting ratings.

We used the *weighted matrix factorization* (WMF) algorithm, proposed by Hu et al. [16], to learn latent factor representations of all users and items in the Taste Profile Subset. This is a modified matrix factorization algorithm aimed at implicit feedback datasets. Let $r_{ui}$ be the play count for user $u$ and song $i$. For each user-item pair, we define a preference variable $p_{ui}$ and a confidence variable $c_{ui}$ ($I(x)$ is the indicator function, $\alpha$ and $\epsilon$ are hyperparameters):

$$
\begin{aligned}
p_{ui} &= I(r_{ui} > 0), & (1) \\
c_{ui} &= 1 + \alpha \log(1 + \epsilon^{-1} r_{ui}). & (2)
\end{aligned}
$$

The preference variable indicates whether user $u$ has ever listened to song $i$. If it is $1$, we will assume the user enjoys the song. The confidence variable measures how certain we are about this particular preference. It is a function of the play count, because songs with higher play counts are more likely to be preferred. If the song has never been played, the confidence variable will have a low value, because this is the least informative case.

The WMF objective function is given by:

$$\min_{x_\star, y_\star} \sum_{u,i} c_{ui}(p_{ui} - x_u^T y_i)^2 + \lambda \left( \sum_u ||x_u||^2 + \sum_i ||y_i||^2 \right), \tag{3}$$

where $\lambda$ is a regularization parameter, $x_u$ is the latent factor vector for user $u$, and $y_i$ is the latent factor vector for song $i$. It consists of a confidence-weighted mean squared error term and an L2 regularization term. Note that the first sum ranges over all users and all songs: contrary to matrix factorization for rating prediction, where terms corresponding to user-item combinations for which no rating is available can be discarded, we have to take all possible combinations into account. As a result, using stochastic gradient descent for optimization is not practical for a dataset of this size. Hu et al. propose an efficient alternating least squares (ALS) optimization method, which we used instead.

## 4   Predicting latent factors from music audio

Predicting latent factors for a given song from the corresponding audio signal is a regression problem. It requires learning a function that maps a time series to a vector of real numbers. We evaluate two methods to achieve this: one follows the conventional approach in MIR by extracting local features from audio signals and aggregating them into a bag-of-words (BoW) representation. Any traditional regression technique can then be used to map this feature representation to the factors. The other method is to use a deep convolutional network.

Latent factor vectors obtained by applying WMF to the available usage data are used as ground truth to train the prediction models. It should be noted that this approach is compatible with any type of latent factor model that is suitable for large implicit feedback datasets. We chose to use WMF because an efficient optimization procedure exists for it.

### 4.1   Bag-of-words representation

Many MIR systems rely on the following feature extraction pipeline to convert music audio signals into a fixed-size representation that can be used as input to a classifier or regressor [5, 17, 18, 19, 20]:

- **Extract MFCCs from the audio signals.** We computed 13 MFCCs from windows of 1024 audio frames, corresponding to 23 ms at a sampling rate of 22050 Hz, and a hop size of 512 samples. We also computed first and second order differences, yielding 39 coefficients in total.
- **Vector quantize the MFCCs.** We learned a dictionary of 4000 elements with the K-means algorithm and assigned all MFCC vectors to the closest mean.
- **Aggregate them into a bag-of-words representation.** For every song, we counted how many times each mean was selected. The resulting vector of counts is a bag-of-words feature representation of the song.

We then reduced the size of this representation using PCA (we kept enough components to retain 95% of the variance) and used linear regression and a multilayer perceptron with 1000 hidden units on top of this to predict latent factors. We also used it as input for the metric learning to rank (MLR) algorithm [21], to learn a similarity metric for content-based recommendation. This was used as a baseline for our music recommendation experiments, which are described in Section 5.2.

### 4.2   Convolutional neural networks

Convolutional neural networks (CNNs) have recently been used to improve on the state of the art in speech recognition and large-scale image classification by a large margin [22, 23]. Three ingredients seem to be central to the success of this approach:

- Using rectified linear units (ReLUs) [24] instead of sigmoid nonlinearities leads to faster convergence and reduces the vanishing gradient problem that plagues traditional neural networks with many layers.
- Parallellization is used to speed up training, so that larger models can be trained in a reasonable amount of time. We used the Theano library [25] to take advantage of GPU acceleration.

- A large amount of training data is required to be able to fit large models with many parameters. The MSD contains enough training data to be able to train large models effectively.

We have also evaluated the use of dropout regularization [26], but this did not yield any significant improvements.

We first extracted an intermediate time-frequency representation from the audio signals to use as input to the network. We used log-compressed mel-spectrograms with 128 components and the same window size and hop size that we used for the MFCCs (1024 and 512 audio frames respectively). The networks were trained on windows of 3 seconds sampled randomly from the audio clips. This was done primarily to speed up training. To predict the latent factors for an entire clip, we averaged over the predictions for consecutive windows.

Convolutional neural networks are especially suited for predicting latent factors from music audio, because they allow for intermediate features to be shared between different factors, and because their hierarchical structure consisting of alternating feature extraction layers and pooling layers allows them to operate on multiple timescales.

### 4.3 Objective functions

Latent factor vectors are real-valued, so the most straightforward objective is to minimize the mean squared error (MSE) of the predictions. Alternatively, we can also continue to minimize the weighted prediction error (WPE) from the WMF objective function. Let $y_i$ be the latent factor vector for song $i$, obtained with WMF, and $y_i'$ the corresponding prediction by the model. The objective functions are then ($\theta$ represents the model parameters):

$$\min_\theta \sum_i ||y_i - y_i'||^2, \qquad (4) \qquad \min_\theta \sum_{u,i} c_{ui}(p_{ui} - x_u^T y_i')^2. \qquad (5)$$

## 5 Experiments

### 5.1 Versatility of the latent factor representation

To demonstrate the versatility of the latent factor vectors, we compared them with audio features in a tag prediction task. Tags can describe a wide range of different aspects of the songs, such as genre, instrumentation, tempo, mood and year of release.

We ran WMF to obtain 50-dimensional latent factor vectors for all 9,330 songs in the subset, and trained a logistic regression model to predict the 50 most popular tags from the Last.fm dataset for each song. We also trained a logistic regression model on a bag-of-words representation of the audio signals, which was first reduced in size using PCA (see Section 4.1). We used 10-fold cross-validation and computed the average area under the ROC curve (AUC) across all tags. This resulted in an average AUC of **0.69365** for audio-based prediction, and **0.86703** for prediction based on the latent factor vectors.

### 5.2 Latent factor prediction: quantitative evaluation

To assess quantitatively how well we can predict latent factors from music audio, we used the predictions from our models for music recommendation. For every user $u$ and for every song $i$ in the test set, we computed the score $x_u^T y_i$, and recommended the songs with the highest scores first. As mentioned before, we also learned a song similarity metric on the bag-of-words representation using metric learning to rank. In this case, scores for a given user are computed by averaging similarity scores across all the songs that the user has listened to.

The following models were used to predict latent factor vectors:

- Linear regression trained on the bag-of-words representation described in Section 4.1.
- A multi-layer perceptron (MLP) trained on the same bag-of-words representation.
- A convolutional neural network trained on log-scaled mel-spectrograms to minimize the mean squared error (MSE) of the predictions.

- The same convolutional neural network, trained to minimize the weighted prediction error (WPE) from the WMF objective instead.

For our initial experiments, we used a subset of the dataset containing only the 9,330 most popular songs, and listening data for only 20,000 users. We used 1,881 songs for testing. For the other experiments, we used all available data: we used all songs that we have usage data for and that we were able to download an audio clip for (382,410 songs and 1 million users in total, 46,728 songs were used for testing).

| Model | mAP | AUC |
|---|---|---|
| MLR | 0.01801 | 0.60608 |
| linear regression | 0.02389 | 0.63518 |
| MLP | 0.02536 | 0.64611 |
| CNN with MSE | 0.05016 | 0.70987 |
| CNN with WPE | 0.04323 | 0.70101 |

Table 2: Results for all considered models on a subset of the dataset containing only the 9,330 most popular songs, and listening data for 20,000 users.

We report the mean average precision (mAP, cut off at 500 recommendations per user) and the area under the ROC curve (AUC) of the predictions. We evaluated all models on the subset, using latent factor vectors with 50 dimensions. We compared the convolutional neural network with linear regression on the bag-of-words representation on the full dataset as well, using latent factor vectors with 400 dimensions. Results are shown in Tables 2 and 3 respectively.

On the subset, predicting the latent factors seems to outperform the metric learning approach. Using an MLP instead of linear regression results in a slight improvement, but the limitation here is clearly the bag-of-words feature representation. Using a convolutional neural network results in another large increase in performance. Most likely this is because the bag-of-words representation does not reflect any kind of temporal structure.

Interestingly, the WPE objective does not result in improved performance. Presumably this is because the weighting causes the importance of the songs to be proportional to their popularity. In other words, the model will be encouraged to predict latent factor vectors for popular songs from the training set very well, at the expense of all other songs.

On the full dataset, the difference between the bag-of-words approach and the convolutional neural network is much more pronounced. Note that we did not train an MLP on this dataset due to the small difference in performance with linear regression on the subset. We also included results for when the latent factor vectors are obtained from usage data. This is an upper bound to what is achievable when predicting them from content. There is a large gap between our best result and this theoretical maximum, but this is to be expected: as we mentioned before, many aspects of the songs that influence user preference cannot possibly be extracted from audio signals only. In particular, we are unable to predict the popularity of the songs, which considerably affects the AUC and mAP scores.

| Model | mAP | AUC |
|---|---|---|
| random | 0.00015 | 0.49935 |
| linear regression | 0.00101 | 0.64522 |
| CNN with MSE | 0.00672 | 0.77192 |
| upper bound | 0.23278 | 0.96070 |

Table 3: Results for linear regression on a bag-of-words representation of the audio signals, and a convolutional neural network trained with the MSE objective, on the full dataset (382,410 songs and 1 million users). Also shown are the scores achieved when the latent factor vectors are randomized, and when they are learned from usage data using WMF (upper bound).

### 5.3 Latent factor prediction: qualitative evaluation

Evaluating recommender systems is a complex matter, and accuracy metrics by themselves do not provide enough insight into whether the recommendations are sound. To establish this, we also performed some qualitative experiments on the subset. For each song, we searched for similar songs by measuring the cosine similarity between the predicted usage patterns. We compared the usage patterns predicted using the latent factors obtained with WMF (50 dimensions), with those using latent factors predicted with a convolutional neural network. A few songs and their closest matches according to both models are shown in Table 4. When the predicted latent factors are used, the matches are mostly different, but the results are quite reasonable in the sense that the matched songs are likely to appeal to the same audience. Furthermore, they seem to be a bit more varied, which is a useful property for recommender systems.

| Query | Most similar tracks (WMF) | Most similar tracks (predicted) |
|---|---|---|
| Jonas Brothers - Hold On | Jonas Brothers - Games<br>Miley Cyrus - G.N.O. (Girl's Night Out)<br>Miley Cyrus - Girls Just Wanna Have Fun<br>Jonas Brothers - Year 3000<br>Jonas Brothers - BB Good | Jonas Brothers - Video Girl<br>Jonas Brothers - Games<br>New Found Glory - My Friends Over You<br>My Chemical Romance - Thank You For The Venom<br>My Chemical Romance - Teenagers |
| Beyoncé - Speechless | Beyoncé - Gift From Virgo<br>Beyonce - Daddy<br>Rihanna / J-Status - Crazy Little Thing Called Love<br>Beyoncé - Dangerously In Love<br>Rihanna - Haunted | Daniel Bedingfield - If You're Not The One<br>Rihanna - Haunted<br>Alejandro Sanz - Siempre Es De Noche<br>Madonna - Miles Away<br>Lil Wayne / Shanell - American Star |
| Coldplay - I Ran Away | Coldplay - Careful Where You Stand<br>Coldplay - The Goldrush<br>Coldplay - X & Y<br>Coldplay - Square One<br>Jonas Brothers - BB Good | Arcade Fire - Keep The Car Running<br>M83 - You Appearing<br>Angus & Julia Stone - Hollywood<br>Bon Iver - Creature Fear<br>Coldplay - The Goldrush |
| Daft Punk - Rock'n Roll | Daft Punk - Short Circuit<br>Daft Punk - Nightvision<br>Daft Punk - Too Long (Gonzales Version)<br>Daft Punk - Aerodynamite<br>Daft Punk - One More Time / Aerodynamic | Boys Noize - Shine Shine<br>Boys Noize - Lava Lava<br>Flying Lotus - Pet Monster Shotglass<br>LCD Soundsystem - One Touch<br>Justice - One Minute To Midnight |

Table 4: A few songs and their closest matches in terms of usage patterns, using latent factors obtained with WMF and using latent factors predicted by a convolutional neural network.

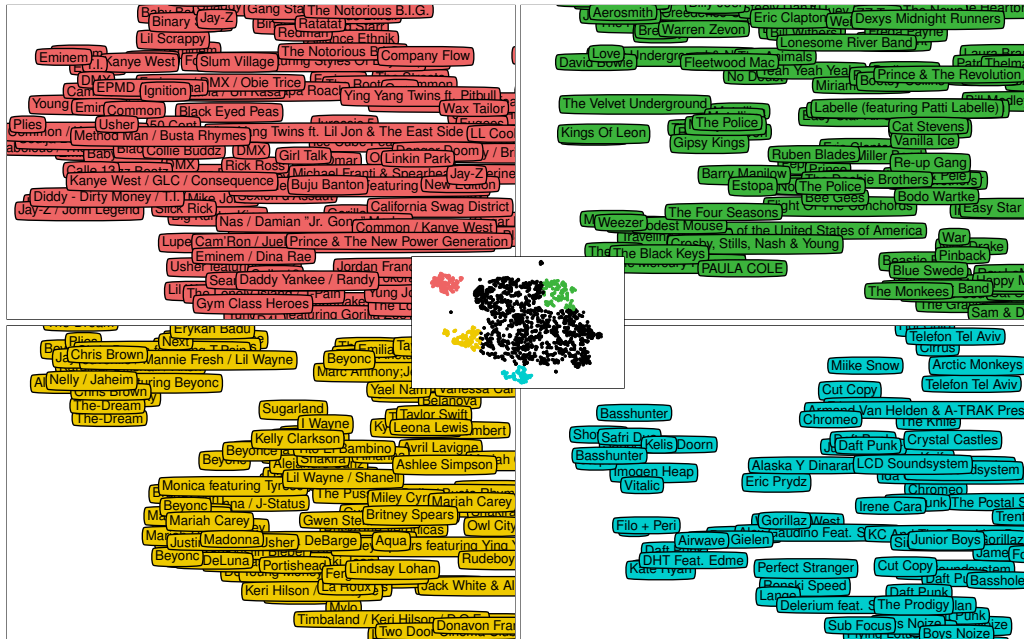

Figure 1: t-SNE visualization of the distribution of predicted usage patterns, using latent factors predicted from audio. A few close-ups show artists whose songs are projected in specific areas. We can discern hip-hop (red), rock (green), pop (yellow) and electronic music (blue). This figure is best viewed in color.

Following McFee et al. [5], we also visualized the distribution of predicted usage patterns in two dimensions using t-SNE [27]. A few close-ups are shown in Figure 1. Clusters of songs that appeal to the same audience seem to be preserved quite well, even though the latent factor vectors for all songs were predicted from audio.

# 6 Related work

Many researchers have attempted to mitigate the cold start problem in collaborative filtering by incorporating content-based features. We review some recent work in this area of research.

Wang et al. [28] extend probabilistic matrix factorization (PMF) [29] with a topic model prior on the latent factor vectors of the items, and apply this model to scientific article recommendation. Topic proportions obtained from the content of the articles are used instead of latent factors when no usage data is available. The entire system is trained jointly, allowing the topic model and the latent space learned by matrix factorization to adapt to each other. Our approach is sequential instead: we first obtain latent factor vectors for songs for which usage data is available, and use these to train a regression model. Because we reduce the incorporation of content information to a regression problem, we are able to use a deep convolutional network.

McFee et al. [5] define an artist-level content-based similarity measure for music learned from a sample of collaborative filter data using metric learning to rank [21]. They use a variation on the typical bag-of-words approach for audio feature extraction (see section 4.1). Their results corroborate that relying on usage data to train the model improves content-based recommendations. For audio data they used the CAL10K dataset, which consists of 10,832 songs, so it is comparable in size to the subset of the MSD that we used for our initial experiments.

Weston et al. [17] investigate the problem of recommending items to a user given another item as a query, which they call 'collaborative retrieval'. They optimize an item scoring function using a ranking loss and describe a variant of their method that allows for content features to be incorporated. They also use the bag-of-words approach to extract audio features and evaluate this method on a large proprietary dataset. They find that combining collaborative filtering and content-based information does not improve the accuracy of the recommendations over collaborative filtering alone.

Both McFee et al. and Weston et al. optimized their models using a ranking loss. We have opted to use quadratic loss functions instead, because we found their optimization to be more easily scalable. Using a ranking loss instead is an interesting direction of future research, although we suspect that this approach may suffer from the same problems as the WPE objective (i.e. popular songs will have an unfair advantage).

## 7  Conclusion

In this paper, we have investigated the use of deep convolutional neural networks to predict latent factors from music audio when they cannot be obtained from usage data. We evaluated the predictions by using them for music recommendation on an industrial-scale dataset. Even though a lot of characteristics of songs that affect user preference cannot be predicted from audio signals, the resulting recommendations seem to be sensible. We can conclude that predicting latent factors from music audio is a viable method for recommending new and unpopular music.

We also showed that recent advances in deep learning translate very well to the music recommendation setting in combination with this approach, with deep convolutional neural networks significantly outperforming a more traditional approach using bag-of-words representations of audio signals. This bag-of-words representation is used very often in MIR, and our results indicate that a lot of research in this domain could benefit significantly from using deep neural networks.

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
