[Reviews · NeurIPS 2013]

Submitted by Assigned_Reviewer_4

The authors practically compare a method using deep convolutional networks (DCN) with the conventional methods using bag-of-words (BOW) representation in the experiments for music recommendation. It is shown that for recommending new and unpopular music which has no usage data available, the method using DCN can be viable. The claim is clear, although it is difficult to find a new methodological concept.

Detailed comments:
- The second line in section 1.1: predictable -> unpredictable
Summary: The claim that for recommending new and unpopular music, the method using DCN can be viable is reasonable, although it is difficult to find a new methodological concept.

Submitted by Assigned_Reviewer_5

The authors demonstrate how to construct a very plausible ground truth dataset for evaluating music recommender systems based on the content of personal music databases, and demonstrate good results for pure content-based recommendation using a combination of convolutional networks and richer audio features than the standard low-dimensional MFCC+VQ feature representation.

The work is of high quality and is very clear indeed. It's very review-like in its tone, taking the time to cover prior work (almost going to extremes in this in fact, leaving perhaps less space than ideal for actual results). Good quality approaches like this, both to construction of believable ground-truth data sets, and in the use of alternative machine learning approaches like CNNs are exactly what the music information retrieval community needs.
Summary: I think this is a solid paper showing interesting results in an emerging field; it's good-quality research, relevant and original. I'm pleased to see the combination of convolutional neural networks and richer audio features lead to such promising results.

Submitted by Assigned_Reviewer_7

This is an interesting paper that addresses a problem of significant practical value. Specifically, the authors propose a technique to learn latent factors directly from music audio without relying on usage data. They do this using deep convolutional neural networks. The paper is well written and the authors do a good job of placing this work in context. The experiments are reasonable. They provide quantitative metrics as well as qualitative results, where appropriate. I particularly like Table 1 and Figure 4.

Using existing machine learning algorithms, the authors provide a nice solution to a hard problem in which there is little previous work. The solution would be very useful in practice. However, there is no algorithmic machine learning contribution. Nevertheless, it could be a good applications paper at NIPS, but it is very clearly an applications paper.
Summary: This is a nice paper that uses existing machine learning ideas to attempt to solve a hard practical problem. it could be a nice applications paper.
Author Feedback

Author rebuttal: First of all, we would like to thank the reviewers for their comments and insights.

We have proposed to cast audio-based content recommendation as a latent factor regression problem using usage data as ground truth for training. We believe this is a novel idea, and by formulating the problem in this fashion we are able to build upon a large pre-existing body of research on regression techniques, as well as latent factor models. As a result, we of course agree with the reviewers that our paper does not contain any purely algorithmic machine learning contributions.

However, we consider the fact that this idea can be applied with any kind of regression technique or latent factor model to be an advantage of our approach. We consciously chose to use existing techniques to be able to demonstrate this. Our motivations for the selection of the specific models and techniques we used are detailed in the paper.


We also agree with the assessment that we have written an applications paper: we set out to solve a particular problem, large-scale content-based music recommendation, taking advantage of recent developments in machine learning and deep learning in particular.

We chose to submit this work to NIPS because we feel that our approach of casting the task as a regression problem opens up a lot of possibilities that deserve the attention of the machine learning community. If we were to send the paper to a conference with a different focus (e.g. music information retrieval or recommender systems), we believe that it would not reach the desired target audience in its entirety. Of course researchers in these domains are part of this audience as well, but we feel that it is particularly important to incite interest in this problem among machine learning researchers. As one of the reviewers remarked, little research has been conducted on this topic so far, especially in a large-scale context. We hope that our paper will stimulate further research in this direction.